# Contemporary Insights into the Biological Mechanisms of *Parkia biglobosa*

**DOI:** 10.3390/ijerph21040394

**Published:** 2024-03-24

**Authors:** Kayode Komolafe, Mary Tolulope Olaleye, Hung-Chung Huang, Maricica Pacurari

**Affiliations:** 1RCMI Center for Environmental Health, College of Science, Engineering and Technology, Jackson State University, 1400 Lynch Street, Box 18750, Jackson, MS 39217, USA; kayode.c.komolafe@jsums.edu; 2Environmental Science PhD Program, College of Science, Engineering and Technology, Jackson State University, 1400 Lynch Street, Jackson, MS 39217, USA; 3Department of Biochemistry, School of Sciences, The Federal University of Technology, P.M.B. 704, Akure 340110, Nigeria; mtolaleye@futa.edu.ng; 4Department of Biology, College of Science, Engineering and Technology, Jackson State University, Jackson, MS 39217, USA; hung-chung.huang@jsums.edu

**Keywords:** *Parkia biglobosa*, extract, ethnomedicine, pharmacological activities, mechanisms

## Abstract

For a long time, traditional medicine has relied on the use of medicinal plants and herbal products which have served as the basis for numerous pharmaceuticals. *Parkia biglobosa* (Jacq) R.Br.ex. G. Don., commonly called the African locust bean tree, is a perennial deciduous plant native to West Africa where it is highly esteemed for its nutritional and traditional medicinal benefits. *Parkia biglobosa*’s ethnomedicinal uses include microbial infections such as diarrhea and chronic diseases like hypertension and type 2 diabetes mellitus. This article presents the current understanding of the molecular mechanisms underlying *Parkia biglobosa*’s biological effects. An electronic database search was conducted using *P. biglobosa* and its synonyms as keywords in Scientific Electronic Library Online, ISI Web of Knowledge, PubMed, Scopus, Science Direct, and Google Scholar. Consistently, scientific research has confirmed the medicinal effects of the plant’s extracts and active phytochemicals, including antimicrobial, analgesic, antidiabetic, antihypertensive, hypolipidemic, and neuroprotective properties, among others. It highlights the contributions of identified innate phytochemicals and existing limitations to therapeutic applications, as well as the need for and prospects for further research. Advancing our understanding of the medicinal plant’s biological mechanisms and the contributions of the active phytochemicals would allow for more effective exploration of its vast pharmacological potential and facilitate clinical applications.

## 1. Introduction

*Parkia biglobosa* (Jacq) R.Br.ex. G. Don. is a perennial deciduous legume that reaches 20–30 m high and belongs to the family Fabaceae. The multipurpose tree, which is commonly known as the “African locust bean tree”, is indigenous to West Africa and can be found in various subtropical African countries. The tree is widely used in the production of food, beverages, medicine, and non-timber forest resources, with its seeds, fruit pulps, and leaves being particularly popular [1,2]. Historically, African locust beans have been largely grown for their seeds, which are fermented to produce a popular condiment used as a seasoning in many African dishes (Figure 1). Similarly, the plant has been widely employed in traditional medicine for centuries to treat a variety of diseases. *Parkia biglobosa* (PB) has more recently aroused the interest of researchers with its promising preclinical findings, suggesting it could be a valuable source of therapeutic compounds. Extracts of the leaf and/or stem bark were reported to exert antioxidant, hypotensive, cardioprotective, antihyperlipidemic, antidiabetic, antiepileptogenic, anti-amnesic, and anxiolytic-like effects in animal models [2,3,4,5,6,7], as well as antidiarrheal and antibacterial properties [8,9]. Furthermore, protein isolates from the seed exhibited hypolipidemic and cardioprotective effects [10,11], while the seeds were found to have vasorelaxant actions on smooth muscles, thereby initiating hypotensive effects [12]. A comprehensive review of the current understanding of the biological activities of *Parkia biglobosa*, including possible signaling pathways and mechanisms affected by the plant, is required in light of the plant’s substantial significance in ethnomedicine and the scientific validation of its medicinal properties. By filling in current knowledge gaps, this paper hopes to lay the groundwork for future research with the ultimate goal of facilitating a more efficient exploration of the plant’s intriguing medicinal properties and the identification of more active medicinal agents.

## 2. Methodology

To begin with, the plant’s name was checked for authenticity using www.theplantlist.org (accessed on 15 February 2023), and synonyms with three confidence levels were considered. *Parkia biglobosa* (Jacq.) G.Don was listed among the 41 accepted species names for the genus *Parkia* in the online plant database, out of a total of 92 scientific plant names [14]. From February to September 2023, the plant name and its synonyms were used in a literature search of the ISI Web of Knowledge, PubMed, Scopus, Science Direct, Google Scholar, and Scientific Electronic Library Online (SciELO), following methods that have already been described [15]. The study databases contained original research papers published in peer-reviewed journals, books, dissertations, and other reports that focused on the *Parkia biglobosa* plant. Retrieved articles detailing plant collection and assays that have been conducted were grouped into ethnomedicinal applications, pharmacology, and phytochemistry. Articles reporting the use of the extract of the plant or phytochemicals isolated or purified from it with proper identification and voucher specimens were considered for this review. As an exclusion criterion, publications from journals or conferences that were deemed unreliable and non-English articles that could not be accessed in full or translated were not considered.

## 3. Results and Discussion

### 3.1. Ethnomedicinal Applications and Phytochemistry

Aside from its nutritional value, *Parkia biglobosa* has long been sought after in traditional medicine. For centuries, different components of the plants have been used to treat a variety of pathological illnesses in Sub-Saharan Africa. Various infusions, decoctions, macerations, and lotions of the seeds, leaves, stem barks, and roots of the plant are typically employed in the treatment of pathological conditions [16], including bacterial infections, malaria, pain, type 2 diabetes mellitus, hypertension, dyslipidemia, and inflammatory diseases [6,16,17,18]. Vapor inhalants, or macerations of the bark are used in the management of ulcers, skin infections, ear pain, toothaches, bronchitis, pneumonia, diarrhea, vomiting, edema, and venereal diseases. The leaves have been used to cure toothaches, burns, and hemorrhoids, while the seed can be used to relieve anxiety. Traditional uses of the pulps include their implementation as mild purgatives, diuretics, and fever treatments, while the roots are used as a lotion to improve eyesight [1,19].

Phytochemical screening data for extracts derived from various parts of *Parkia biglobosa* are abundant in the scientific literature. Polyphenols, alkaloids, cardiac glycosides, saponins, terpenoids, and steroids have all been identified in various extracts of the leaf, seed, root, and bark of the medicinal plant. In several studies, the bioactive molecules of PB were isolated and purified via bioactivity-guided solvent extraction and thin-layer, column, and high-performance liquid chromatographic techniques, while its structures were characterized using gas chromatography coupled with electrospray ionization mass spectrometry (GC-ESI/MS) [2,20,21,22,23,24]. Phenolic acids like gallic acid, caffeic acid, chlorogenic acid, and flavonoids/tannins including quercetin, rutin, epicatechin, epigallocatechin, epigallocatechin gallate, isorhamnetin-3-*O*-rutinoside, naringenin 7-4’-di-*O*-β-d-glucopyranoside, 1-(ω-feruloyllignoceryl)–glycerol, and 4-*O*-methyl-epigallocatechin were all identified in the leaf, root, and bark [2,23,24,25,26]. Following solvent extraction of PB seed oil and TLC separation of the sterol fractions, the GC-MS technique was used to determine the individual sterols, including b-sitosterol, stigmasterol, campesterol, and Δ^5^-avenasterol [20,21]. Lastly, terpenoids like limonene and lupeol and fatty acids including methyl-6-cis-9-11-trans octadecatrienoate, linoleic acid, and docosanoic acid were also characterized [20,21,26,27] from various parts of the plant, as shown in Table 1.

**Table 1 ijerph-21-00394-t001:** Some chemical compounds identified, isolated, or characterized from *Parkia biglobosa*.

Chemical Class	Compound Name	Plant Part	Extraction Method			Isolation/Analytical Technique				Reference
Phenols	2-methoxy phenol	Seed	Simultaneous distillation–extraction (SDE)			GC-MS				[22]
	2-methoxy-4-methyl-phenol									
	4-ethyl-2-methoxy phenol									
	2,4-disiopropyl-phenol									
Phenol/Phenolic acids	Caffeic acid	Leaf	Aqueous–methanol maceration			HPLC				[28]
	Gallic acid	Leaf								
	Chorogenic acid	Leaf								
Flavonoids/tannins	Catechin Epicatechin Epigallocatechin Epigallocatechin gallate Epigallocatechin-*O*-glucuronide Epicatechin-*O*-gallate-*O*-glucuronide Epigallocatechin-*O*-gallate-*O*-glucuronide Rutin Quercetin Kaempferol Naringenin 7-4’-di-*O*-β-d-glucopyranoside Methoxyluteolin-7-*O*-rutinoside 1-(ω-Feruloyllignoceryl) –glycerol 1-(ω-Isoferuloylalkanoyl) –glycerols 4-*O*-Methyl-epigallocatechin	Leaf, bark Leaf, bark, root Leaf, bark, root Leaf, bark, root Bark, root Bark, root Bark, root Fruit pulp Stem bark Bark Bark Bark	Water, aqueous–methanol, methanol, dichloromethane–methanol, ethyl acetate, butanol, hexane, and chloroform maceration and/or fractionation			Column chromatography, TLC, HPLC, mass spectrometry, HPLC/ESI-IT-MS, FIA-ESI-IT-MS, LC-HR-QTOF, and NMR				[23,25,26,28] [24]
Phytosterols	*b*-Sitosterol	Seed	N-hexane, diethyl ether percolation and maceration			TLC, gas chromatography, and GC-MS				[20,21]
	Campesterol	Seed								
	Stigmasterol	Seed								
	Δ^5^-Avenasterol	Seed								
	Δ^7^-Stigmasterol	Seed								
	Cholesterol	Seed								
Terpenoids	Limonene	Seed	Ethylacetate maceration,			SDE				[22,26]
	Lupeol	Bark								
Lipid/Fatty acids	Dodecanoic acid	Stem bark	N-butanol, N-hexane, and petroleum ether maceration and percolation			TLC, GC-FID, and GC-MS				[20,27,29,30,31]
	1,2,3-propanetriol	Stem bark								
	Arachidic acid	Seed								
	Arachidonic acid	Seed								
	Methyl-6-cis-9-11-trans-octadecatrienoate Palmitic acid	Seed								
	Stearic acid	Seed								
	Oleic acid	Seed								
	Linoleic acid	Seed								
	Linolenic acid	Seed								
	Docosanoic acid	Seed								

ESI: Electrospray ionization; FIA: Flow injection analysis; GC-FID-Gas chromatography with flame ionization detection; GC-MS: Gas chromatography-mass spectrometry; HPLC: High-performance liquid chromatography; IT: Ion trap; LC-HR-QTOF: liquid chromatography high-resolution quadrupole time-of-flight mass spectrometry; MS: Mass spectrometry; NMR: Nuclear Magnetic Resonance spectroscopy; QToFMS: Quadrupole time-of-flight mass spectrometry; SDE: Simultaneous distillation-extraction; TLC: Thin-layer chromatography.

### 3.2. Pharmacological Activities and Biochemical Signaling

Scientific research on the numerous molecular signaling pathways connected to the pharmacological actions of *Parkia biglobosa* is still in its nascent stages. However, research has shown following signaling pathways are involved in mediating *Parkia biglobosa* effects: antioxidant, anti-inflammatory, lipid, glucose metabolism, renin–angiotensin, and nitric oxide pathways (Table 2). Here, we discuss the current knowledge about the plant’s pharmacological activities and the biochemical processes underlying these effects.

#### 3.2.1. Cardioprotective and Antihypertensive Properties

Studies have shown that *Parkia biglobosa* has therapeutic effects on cardiovascular disorders. Several plant parts, such as the seed [10,11,12], stem bark [32], and leaf [5,7], have been demonstrated by both empirical and scientific evidence to possess cardioprotective and hypotensive properties. Potential therapeutic mechanisms through which *Parkia biglobosa* exerts its effects include the augmentation of nitric oxide (NO) bioavailability and enhancement of endothelial function, both of which promote antioxidant and anti-inflammatory activities. Additionally, the plant facilitates arterial vasodilation and regulates the renin–angiotensin–aldosterone system (RAAS) [28,33]. 

##### Cardioprotective Mechanisms

Antioxidant, Anti-Inflammatory, and Antihyperlipidemic Mechanisms

The hydroalcoholic extracts of *Parkia biglobosa* leaf and stem bark offer cardioprotective benefits against doxorubicin- and isoproterenol-induced myocardial infarction in rats, respectively [5,34]. The antioxidant and antihyperlipidemic actions of the leaf’s phytochemicals, particularly polyphenolics, were suggested to contribute to the amelioration of doxorubicin-induced cardiotoxicity in rats [5]. Saponins from *Panax quinquefolium*, for example, were responsible for improving neonatal rat cardiomyocyte viability as well as the plant’s cardioprotective benefits in an animal model of myocardial infarction [35]. The therapeutic properties of *Parkia biglobosa* plant parts, such as its ability to lower blood pressure, are largely attributed to the presence of antioxidant phytochemicals, specifically ascorbic acid and polyphenols and their derivatives (flavonoids, procyanidins, phenolic acids, and tannins) [6,36]. Lines of evidence support that polyphenolic substances participate in metabolic events that prevent or mitigate cardiovascular diseases, including hypertension, and flavonoids constitute the most characterized and quantitatively most important subset of polyphenols [37]. Additionally, an aqueous extract of *Parkia biglobosa* stem bark was found to ameliorate some risk markers of cardiometabolic diseases caused by high-salt diet ingestion in rats via its antilipidemic, anti-inflammatory, and uricosuric effects, even though the individual contributions of the phytochemicals to the observed effects were not elucidated [30]. The observation of an increased HDL-C/LDL-C ratio in diabetic animals fed an aqueous extract of fermented *Parkia biglobosa* seed led to the supposition that the seed could protect against the development of cardiovascular diseases in individuals at risk [38]. It was found that protein isolates from *Parkia biglobosa* seeds could protect the hearts of diabetic rats by lowering blood sugar, triglycerides, and oxidative stress in the heart [11].

Vasorelaxation and Endothelium-Dependent Nitric Oxide (ENOS) Activity

Hypertension can be linked to increased vascular constriction caused by a decrease in the production or release of substances that relax the blood vessels, such as nitric oxide (NO) and endothelium-derived hyperpolarizing factor (EDHF). Certain sources rich in polyphenols, like grape juice and red wine, have been found to have therapeutic effects by improving these processes [7]. The procyanidin polyphenols in *Parkia biglobosa* leaf were found to induce redox-sensitive, endothelial-dependent vasorelaxation [7]. Such polyphenol-dependent vasodilation is mediated by stimulation of the Src/PI3-kinase/Akt pathway, which enhances endothelial NO synthase activity [39]. *Parkia biglobosa* has an array of active compounds, many of which have been shown to protect against cardiovascular disease. Quercetin was found to be protective in stage 1 hypertensive humans [40] and could increase aortic eNOS activity in obese Zucker rats, thus improving endothelial function and causing up to a 72% decrease in systolic blood pressure compared to the control [41]. Kaempferol, identified in the leaf of *Parkia biglobosa* [2], was found to reduce the MAP of adult female rats at a high dose (10 mg/kg) due to its efficiency as an endothelium-independent vasodilator [42]. Similarly, epicatechin, a flavanal found in the bark and leaves of *Parkia biglobosa*, has been shown to lower systolic blood pressure in spontaneously hypertensive rats by increasing aortic eNOS activity and NO availability while facilitating acetylcholine-mediated vasodilation [43].

Inhibition of Angiotensin-Converting Enzyme (ACE)

Inhibiting the angiotensin-converting enzyme (ACE) is one mechanism by which many medicinal plants exert their hypotensive effects. Angiotensin II, a powerful vasoconstrictive and hypertensive agent, is synthesized by the ACE. *Parkia biglobosa* leaf extract has been linked to ACE-inhibitory activity due to its active polyphenolic components, including catechins [28]. In one study, the hypotensive effect of green tea in diabetic Goto-Kakizaki (GK) rats was linked to the ACE-inhibitory effects of flavanol catechins such as epigallocatechin gallate (EGCg), epicatechin gallate (ECg), and epicatechin [44]. One possible mechanism for phenolic inhibition of the ACE is the formation of chelates capable of building complexes within the active center of ACE and thereby inactivating the enzyme [45]. Komolafe et al. [28] posited that the interaction of one or more of the *Parkia biglobosa* leaf’s constituent phenolics with the ACE is responsible, at least in part, for the previously reported hypotensive effect of the leaf [6].

#### 3.2.2. Antidiabetic and Hypolipidemic Effects

The scientific literature extensively documents the hypoglycemic and hypolipidemic characteristics of *Parkia biglobosa*. An aqueous extract of fermented PB seeds was found to effectively reduce the symptoms of diabetes in rats induced with alloxan [38]. This extract was found to reverse high blood sugar levels, abnormal lipid levels, weight loss caused by the breakdown of structural proteins, and muscle wasting. The effects of the extract were comparable to those of glibenclamide, a commonly used medication for diabetes. In addition, hypoglycemic and hypolipidemic effects were observed in normoglycemic rodents in response to PB extracts, while hyperglycemia triggered by alloxan and glucose in mice was reversed [46,47]. 

##### Antidiabetic Mechanisms

Insulin-like Function, Antioxidant/Anti-Inflammatory Effects, and Modulation of Carbohydrate Metabolizing Enzymes

There are indications that the antidiabetic actions of *Parkia biglobosa* involve multiple mechanisms; however, further research is needed to identify the active molecules and understand their specific roles in these effects. Ibrahim et al. [3] reported that the butanol component of PB leaf was effective in reducing high blood sugar levels and managing diabetic complications in rats with type 2 diabetes. This effect is likely due to its ability to enhance the function of pancreatic β-cells and increase insulin secretion. The reduction in diabetic hyperlipidemia achieved by using the isolate from PB seeds was also linked to the insulin-like proteins and insulin-releasing properties of certain active components found in the seeds [48], since insulin can lower lipid levels in diabetic animals [49]. Ogunyinka et al. [11] found that the protein isolates from PB seeds had a dosage-dependent antihyperglycemic impact in diabetic rats induced by STZ. This effect was comparable to that of insulin at a dose of 5 U/kg. The researchers discovered that the seed isolate stimulated the synthesis of natural antioxidants within the body and inhibited oxidative stress caused by hyperglycemia. Therefore, they deduced that the seed isolate can enhance antioxidative processes, which could lead to an improvement in diabetic symptoms and a decrease in the likelihood of diabetes-related complications. *Parkia biglobosa* possesses an impressive antioxidant profile, with the leaf demonstrating the most significant proportion of radical scavenging activity compared to two other medicinal plants. The correlation between this phenomenon and the comparatively elevated levels of polyphenolic compounds in the leaf, as opposed to other plant infusions, was established [50]. The authors of the study hypothesized that the antihyperglycemic effect of PB seed and the improvement in abnormal biochemical and hematological markers in rats induced with STZ may be attributed, at least partially, to the presence of anti-inflammatory and antioxidant phytochemicals in the seed [51]. The hypoglycemic effect of the PB leaf may be attributed to its ability to inhibit key enzymes involved in glucose metabolism, such as α-amylase and α-glucosidase [24,52]. The hypoglycemic impact of *Parkia biglobosa* was linked to the inhibitory effects of a triterpenoid ingredient, lupeol, on α-amylase and α-glucosidase enzymes [3]. Tamfu et al. [53] also discovered that the in vitro inhibitory effectiveness of PB stem bark extracts against both metabolic enzymes was superior compared to that of the reference medication, acarbose.

#### 3.2.3. Antimicrobial Activity of Parkia biglobosa

There is a wealth of literature that supports the antibacterial properties of *Parkia biglobosa*. Bactericidal properties were reported for a crude methanol extract of *Parkia biglobosa* stem bark as well as its aqueous and n-hexane fractions [19]. The extracts’ phytocomponents inhibited the growth of 15 Gram-positive and Gram-negative bacterial isolates. The extract’s zones of inhibition were between 14 0.00 mm and 28 0.71 mm, which compared favorably to the effect of the reference antibiotic, streptomycin (0 0.00 mm and 24 0.71 mm) [19]. Eboma et al. [54] demonstrated the antibacterial activity of the aqueous extract and oil of fermented locust bean seeds against 14 common pathogenic microorganisms and asserted that the oil had a higher efficiency, as indicated by a larger zone of inhibition. As with the aqueous extracts of *Parkia biglobosa* root [55] and stem bark [56], the seed aqueous extract inhibited methicillin-resistant *S. aureus* (MRSA) at doses ranging from 5 to 100 mg/mL [54]. Experimental rats infected with the bacteria and treated with the aqueous or n-hexane extracts of the fermented seeds showed an improved white blood cell count and packed cell volume (PCV). Furthermore, aqueous and acetone extracts of fermented and unfermented *Parkia biglobosa* seeds inhibited clinical isolates including *P. aeruginosa*, *E. coli*, and *Candida albicans* [57], while *Parkia biglobosa* aqueous root and leaf extracts inhibited *S. aureus* and *E. coli* [58]. In the same vein, a methanolic extract of the stem bark of *Parkia biglobosa* proved effective against castor oil- and E. coli-induced diarrhea in rats by increasing the frequency and consistency of bowel movements, restoring villi integrity, and lowering the fecal microbial load and goblet cell hyperplasia brought on by mucus buildup in the animals’ droppings [59]. Similar to a more recent discovery [60], the leaf and seed extracts were found to be efficacious against *B. cereus, S. aureus,* and *E. coli*. In terms of fungicidal activity, aqueous and ethanolic extracts of *Parkia biglobosa* leaf and bark demonstrated concentration-dependent, growth-inhibitory activity against *Candida albicans*, the most common infection-causing fungus, with the water extract producing a superior zone of inhibition and a lower minimum inhibitory concentration (MIC) [61].

##### Mechanisms of Antimicrobial Effects

Cytotoxic Activities of Essential Oils and Polyphenols

Although it is acknowledged that the antibacterial action of *Parkia biglobosa* can be attributed to the constituent saponins, tannins, glycosides, and alkaloids [19,57], very little research has been done to provide information on the mechanism(s) involved. Eboma et al. [54] submitted that bioactive compounds in fermented *Parkia biglobosa* seed oil, such as ricinoleic acid, p-cymene, octadecanoic acid, and n-hexadecanoic acid, were partially responsible for the antibacterial action. Some of these compounds, like p-cymene and ricinoleic acid, are known anti-inflammatory, antinociceptive, anxiolytic, and antimicrobial agents [62]. Bioactive components of essential oils might perturb the structural integrity of the microbial cell membrane bilayer, with consequential effects on cellular metabolism and eventual death [63]. Ricinoleic acid, palmitic acid, and stearic acid, which are found in the seed oil of *Parkia biglobosa*, have been shown to exert antibacterial properties primarily by altering the fluidity of cellular membranes and disrupting the functioning of crucial membrane proteins [64]. Banwo et al. [59] stated that *Parkia* biglobosa’s antibacterial activity may be due to its high polyphenolic contents, which is in line with a previous claim [65]. The antibacterial activity of polyphenols is well-established and has been linked to their damaging interactions with bacterial cell surfaces [66]. Polyphenols, including flavanols (e.g., epicatechin, catechin, and procyanidins), flavonols (e.g., quercetin and kaempferol), and phenolic acids (e.g., caffeic acid and gallic acid), have repeatedly been shown to be effective against pathogenic microbes. These compounds possess structural characteristics, such as the chalcone skeleton present in certain flavonoids, which may be chemically modified to confer antimicrobial properties [67]. Natural phenolic compounds act through multiple mechanisms, including hindering microbial cell wall, protein, and nucleic acid synthesis; disrupting metabolic pathways; and compromising cell membrane integrity. They can hinder bacterial virulence factors, suppress the formation of biofilms, and even enhance the effectiveness of antibiotics [67,68]. It should be noted, however, that the health impacts of polyphenols are contingent upon their absorption efficiency as well as their levels of ingestion. The bioavailability of polyphenolic compounds is determined by several factors, including intestinal absorption, metabolism by gastrointestinal flora, and urinary excretion. Isoflavones and gallic acid are the polyphenols that exhibit the highest absorption efficiency in humans. Catechins, flavanones (i.e., naringenin), and quercetin glucosides subsequently follow with varying kinetics of absorption [69].

#### 3.2.4. Antimalarial (Antiplasmodial), Antipyretic, and Analgesic Activities

Crude leaf extract and phenolic extract of *Parkia biglobosa* stem bark reportedly showed an antiplasmodial effect against the clinical isolate of P. falciparum in vitro and reduced the mean parasite count in chloroquine-sensitive, *P. berghei*-induced mice in a dose-dependent manner [70]. In a model of yeast-induced pyrexia in which fever was induced via increased prostaglandin synthesis, the administration of methanol extract (100 mg/kg p.o.) and a solvent fraction (50 mg/kg) of PB stem bark to experimental rats resulted in a significant reduction in elevated temperature compared to acetaminophen [71]. Correspondingly, hexane extract of the stem bark had a noticeable analgesic effect and a significant anti-inflammatory effect on mice when used to treat the acetic-acid-induced abdominal writhing test and the carrageenan-induced rat paw edema model, respectively [72].

##### Mechanisms of Antimalarial and Analgesic effects

Disruption of Redox Homeostasis

Although the precise mechanisms by which *Parkia biglobosa* combats malaria are still unresolved, findings have suggested that the plant’s phenolic components could play a pivotal role in this effect [70]. It is hypothesized that the phenolic moieties in phenolic compounds may have antimalarial effects by inhibiting the breakdown of hemoglobin and the formation of hemozoin, preventing the redox homeostasis required for the parasites to thrive, limiting the parasite’s ability to cause oxidative damage, and chelating the iron required for the parasites’ metabolism [73]. Though there is no molecular elucidation of the analgesic effect, the effect could involve peripheral mechanisms since the extract, like acetaminophen, was not sensitive to the hot plate test [72].

#### 3.2.5. Neuroprotective Activity

*Parkia biglobosa* exhibits neuroprotective qualities, according to a handful of studies. In rat brain mitochondrial preparations, it was found to modulate Na^+^ / K^+^-ATPase activity and improve ROS formation [2]. More recently, an aqueous extract of PB stem bark was found to protect mice from pentylenetetrazole-induced epileptic seizures, amnesia, and anxiety-like behavior while also reducing oxidative stress, inflammation, and degenerated or necrotic neurons in the hippocampus [4]. In vitro, *Parkia biglobosa* bark extract conferred protection against the cytotoxic, neurotoxic, and hemotoxic effects of venoms from two lethal snakes, *Naja nigricollis* and *Echis ocellatus*, and was able to delay mortality from the venoms in mice under in vivo conditions [74].

##### Mechanisms of Neuroprotection

GABAergic, Antioxidant, Anti-inflammatory Activity, Protein Precipitation, and Chelation

The neuroprotective properties of *Parkia biglobosa* are thought to be attributed, at least partially, to its GABAergic, antioxidant, and anti-inflammatory signaling mechanisms [4]. Several studies confirmed the antioxidant and anti-inflammatory potentials of the *Parkia biglobosa* plant in both in vitro and in vivo settings, and some of the therapeutic efficacy of the plant proceeded via these mechanisms [4,75]. For example, the protective effect of *Parkia biglobosa* leaf extract on rat hippocampal tissue damage caused by neurotoxicants was partially achieved through the action of antioxidant phenolic components [2]. An anti-inflammatory evaluation of the tincture of the stalk revealed dose-dependent inhibition of croton oil ear inflammation and paw edema induced by both carrageenan and arachidonic acid in a study motivated by the ethnomedicinal use of *Parkia biglobosa* stalk in the treatment of arthritic pains [75]. The underlying basis of the stalk’s anti-inflammatory effects was linked to the inhibition of lipoxygenase, cyclooxygenase, or both of these pathways, based on a comparison of its impact with that of the dual-blocker phenidone [75].

**Table 2 ijerph-21-00394-t002:** Biochemical mechanisms and active constituents underlying the biological effects of *Parkia biglobosa*.

Biological Effects	Part Used	Processing	Active Component or Phytochemicals	Dosages/ Concentrations	EC_50_/IC_50_ or MIC Values	Major Findings	Mechanisms Implicated	Refs
Cardioprotective Activity								
Hypotensive	Seed	Fermented condiments	Not described	100 g/day (in meal)	NA	Individuals known to traditionally consume condiments containing fermented *Parkia biglobosa* seed had significantly lower systolic and diastolic blood pressures, TC/HDL-C ratios, and LDL-C/HDL-C ratios when compared to the group that never consumed the seed	Antihyperlipidemic, antiatherogenic effects	[10]
Vasorelaxant, antihypertensive	Seed (roasted, fermented)	Aqueous extract	Not described	Up to 10 mg/mL	Fermented (EC_50_): 5.37 ± 0.12 and 4.19 ± 1.02 mg/mL Roasted (EC_50_): 5.39 ± 1.12 and 5.93 ± 0.95 mg/mL	Seed extract caused concentration-dependent relaxation of intact and endothelium-denuded aorta of rats subjected to contraction through phenylepinephrine stimulation	Direct action on the smooth muscle and generation of vasodilatory prostaglandins like PGI_2_	[12]
Vasorelaxant	Leaf	Hydroalcoholic extract, procyanidin-rich fractions	Procyanidins, polyphenolics	0.0001–0.1 g/L	NA	In porcine coronary artery rings previously contracted with U46619, *Parkia biglobosa* extract and phenol-rich fractions elicited redox-sensitive, endothelium-dependent relaxations	Induction of redox-sensitive, endothelium-derived relaxing factors like NO and EDHF using the procyanidin polyphenols; promotion of endothelial NO synthase activity via activation of the Src/PI3-kinase/Akt pathway	[7]
Angiotensin-converting enzyme inhibition	Leaf	Phenol-rich extract	Polyphenols	10, 25, and 50 μg/mL	Free phenolics: IC_50_ = 15.35 ± 4.0 μg/mL Bound phenolics: IC_50_ = 46.85 ± 3.3 μg/mL)	Treatment using ACE preparation derived from rat lung with free phenolics of *Parkia biglobosa* caused inhibition of the activity of the vasoconstrictor	Probable chelation of the zinc ions in the ACE active site via free hydroxyl groups of phenolic compounds in *Parkia biglobosa* leaf	[28]
Cardioprotective	Seed (raw, fermented)	Protein-rich extract	Protein isolates	200 and 400 mg/kg,	NA	Protein isolates from fermented PB seeds mitigated diabetes-induced dyslipidemia and cardiac oxidative stress in rats	Stimulation of endogenous enzymatic and non-enzymatic antioxidant production and suppression of mitochondrial ROS generation	[11]
Cardioprotective	Leaf	Hydromethanol extract	Not described	25, 50, 75, and 100 mg/kg, p.o.	NA	PB leaf extract ameliorated doxorubicin-induced cardiac damage, aberrant lipid profile, and oxidative stress	Possible antioxidant and antihyperlipdemic mechanisms	[5]
Cardioprotective	Stem bark	Hydroethanol extract	Not described	60 and 90 mg/kg p.o		*Parkia biglobosa* leaf extract mitigated isoproterenol-induced hypertrophy, cardiac injury, and oxidative stress	Unknown	[34]
Antidiabetic and hypolipidemic activities								
Antidiabetic, antihyperlipidemic	Seed (fermented)	Aqueous and methanol extract	Not described	6 g/kg	NA	Aqueous extract of fermented *Parkia biglobosa* seed exhibited antihyperglycemic effect and increased HDL-C/LDL-C ratio in alloxan-induced diabetic rats	Unknown	[38]
Antidiabetic	Seed	Aqueous extract	Not described	200, 400, and 800 mg/kg p.o	NA	PB seed corrected hyperglycemia and derangements in diabetes-induced hematological, biochemical, and tissue histological markers in stimulation of b-cell function and insulin secretion and inhibition of carbohydrate-metabolizing enzyme STZ-induced rats	Unclear; possible involvement of antioxidant and anti-inflammatory signaling	[51]
Antidiabetic	Leaf	Butanol fraction	Lupeol	150 and 300 mg/kg b.w. p.o	NA	Extract decreased antioxidant actions and release of insulin and insulin-like proteins; postprandial blood glucose level and diabetic complications in type 2 diabetic rats; inhibited α-amylase and α-glucosidase activities in vitro	Stimulation of b-cell function and insulin secretion and inhibition of carbohydrate-metabolizing enzymes	[3]
Antidiabetic, antihyperlipidemic	Seed	Protein-rich extract	Protein isolates	200 and 400 mg/kg b.w.	NA	*Parkia biglobosa* seed protein reversed hyperglycemia and hyperlipidemia in STZ-induced diabetic rats	Antioxidant actions and release of insulin and insulin-like proteins	[11]
Hypoglycemic, hypolipidemic	Seed (fermented)	Methanol extract	Not described	3 mL/100 g b.w. p.o.	NA	Extract of fermented *Parkia biglobosa* seed caused hypoglycemic and hypolipidemic effects in normoglycemic rats	Not described	[46]
Antimicrobial activity								
Antibacterial	Stem bark	Methanol extract; aqueous and n-hexane fractions	Not characterized	Up to 20 mg/mL	Methanol extract: MIC = 0.63–5 mg/mL) Aqueous n-hexane fractions: MIC = 0.63–10 mg/mL)	Stem bark extract and fractions inhibited the growth of selected Gram-positive and Gram-negative bacterial isolates	Not described	[19]
Antibacterial, antifungal	Seed (fermented)	Oil and aqueous extract	Phenolic compounds, p-cymene, fatty acids	2.5–10.0 mg/mL	Oil: MIC = 2.5–10.0 mg/mL Aqueous extract: MIC = 5.0–10 mg/mL	Aqueous extract and oil of fermented *Parkia biglobosa* seed exerted growth-inhibitory effects on selected bacterial and fungal clinical isolates	Not described	[54]
Antibacterial, antifungal	Seed (raw, fermented)	Aqueous and acetone extracts	Not described	75 and 100 mg/mL	Aqueous extract: MIC = 50–75 mg/mL) Acetone extract: MIC = 50–100 mg/mL	Aqueous and acetone extracts of fermented *Parkia biglobosa* exhibited bactericidal/fungicidal effects against *P. aeruginosa*, *E.coli,* and *C. albicans*	Not described	[57]
Antibacterial, antidiarrheal	Stem bark	Methanol extract		37.5–600 mg/mL	NA	Extract protected against diarrheagenic *E. coli*- and castor oil-induced diarrhea in rats	Not fully elucidated; could involve the antioxidant action of polyphenols	[59]
Antimalarial (antiplasmodial), antipyretic, and analgesic activities								
Antiplasmodial	Leaf, stem bark	Decoction, phenol extract	Not characterized	In vivo: 10, 20, and 30 mg/kg In vitro: 1.6, 3.1, and 6.3 µg/mL	Phenol extract: IC50 = 0.51 µg/mL	Extract caused dose-dependent reduction of parasitemia in *P. berghei*-infected mice; exhibited in vitro cytotoxic effect against *P. falciparum*	Not fully elucidated; could involve antioxidant mechanisms	[70]
Antipyretic	Stem bark	Methanol extract and solvent fractions	Not characterized	25, 50, and 100 mg/kg, p.o.	NA	Extract and fractions caused dose-dependent inhibition of yeast-induced pyrexia in albino rats	Mechanism not elucidated; possible influence on prostaglandin synthesis.	[71]
Analgesic, anti-inflammatory	Stem bark	Hexane extract	Partially characterized, sterols, fatty acids	Analgesic: 50, 100, and 200 mg/kg, p.o. or 6.25, 12.5, and 25 mg/kg, i.p. Anti-inflammatory: 100, 200, and 400 mg/kg p.o or 25, 50, and 100 µg topical	Analgesic effect: ID_50_ (p.o.) = 81.5 mg/kg ID_50_ (i.p.) = 12.5 mg/kg Anti-inflammatory effect: ID_50_ (p.o.) = 248.8 mg/kg ID_50_ (topical) = 31.7 µg	Marked inhibition of acetic-acid-induced writhing in mice following intraperitoneal administration of the extract. Protection against phorbol myristate acetate (PMA)-induced ear oedema in mice.	Peripheral mechanisms of pain inhibition; anti-inflammatory mechanisms	[72]
Neuroprotective Activity								
Anti-epileptogenic, anti-amnesic, anxiolytic	Stem bark	Decoction	Not characterized	80, 160, and 320 mg/kg, p.o.	NA	Protection against pentylenetetrazole-induced seizures and hippocampal necrosis; mitigation of anxiety and memory impairment in mice	Elevation in γ-aminobutyric acid (GABA) level and reduction in oxidative stress and inflammatory events	[4]
Neuromodulatory	Leaf	Hydroalcoholic extract	Antioxidant Polyphenols, catechin	25, 50, 100, or 200 µg/mL	NA	Protection of rat hippocampal slices against toxicant-induced damage; increased cerebral Na/K ATPase activity	Antioxidant mechanisms; mild mitochondrial membrane depolarization	[2]
Anticancer effect								
Chemocytotoxic	Leaf	Methanol extract	Not characterized	0.01 μg/mL–200 μg/mL	56.1–136.0 µg/mL	Extract showed *in vitro* cytotoxic effect against selected human breast, colon, and prostate cancer cell lines	Not clarified	[76]
Chemocytotoxic	Seed	Fermentation	Not characterized	0.5–2.5 mg/mL	IC_50_ values 24, 48, and 72 h treatment = HepG2: 1.3, 1.2, and 0.8 mg/mL HeLa: 0.6, 0.4, and 0.3 mg/mL	Extract of fermented seed showed selective cytotoxicity against human hepatocellular (Hep-G2) and cervical (HeLa) cancer and non-cancer cell lines	Possible proapoptotic processes promoted by the bioactive peptides	[77,78]

The biochemical basis through which *Parkia biglobosa* bark offered protection against snake venom was, however, not reported by Asuzu and Harvey [74]. It is established, however, that herbal antivenoms work through a variety of mechanisms, including protein precipitation, enzyme inhibition or inactivation, and chelation via active phytoconstituents such as polyphenolics [79].

#### 3.2.6. Anticancer Activity

Despite the paucity of research on the anticancer properties of *Parkia biglobosa*, some data indicate that the plant’s components may have cytotoxic effects on certain cancer cell lines when tested in vitro. In an in vitro cytotoxic evaluation of some selected Nigerian medicinal plants, the methanol extract of *Parkia biglobosa* leaf was found to be active against human breast adenocarcinoma BT-549 and BT-20, prostate adenocarcinoma PC-3, and colon adenocarcinoma SW-480 cells, with IC_50_ values ranging from 56 to 136 μg/mL [76]. At a 200 µg/mL concentration, *Parkia biglobosa* extract caused a 75%, 72%, and 93% inhibition of BT-549, BT-20, and PC-3, respectively, when compared with the vehicle-treated control cells [76]. Also, an aqueous extract of fermented *Parkia biglobosa* seed condiment demonstrated selective and progressive cytotoxicity against MCF-7, a cell line from human breast epithelial adenocarcinoma, with an IC_50_ of 0.98 mg/mL as opposed to normal human fibroblast KMST-6 (IC_50_ = 1.37 mg/mL) following 48 h of treatment [78]. Similarly, the fermented seed was shown to have cytotoxic effects on both HepG2 and HeLa cells.

##### Mechanisms of Anticancer Effect

Proapoptotic Signaling

The putative anticancer properties of *Parkia biglobosa* seeds may be attributed to the proapoptotic mechanisms facilitated via the bioactive peptides, which are generated by the proteolytic *Bacillus* spp. and are particularly abundant during the fermentation of *Parkia biglobosa* seeds [77]. This aligns with the discovery that kefir, a fermented milk product, derives some of its anticancer properties from the proapoptotic effects of its active peptides [80].

## 4. Limitation and Future Prospect

Despite the promising aspects of *Parkia biglobosa* from ethnomedicinal and empirical viewpoints, there are still significant limitations hampering its utilization for disease management. To date, research on the medicinal plant mostly focuses on the crude extract and partially characterized isolates. Further investigation is required to identify the active components in the plant and correlate them to their biological effects. One significant drawback of utilizing crude extracts is the influence of soil, climate, and harvest circumstances on efficacy, which complicates the capacity to carry out experiments under controlled conditions with reproducible results. Enhanced reproducibility of outcomes is ensured when studying dose-response or cause-effect relationships with characterized and standardized active components rather than crude extracts or isolates containing multiple phytochemicals that may interact additively, synergistically, or antagonistically. Such standardization will enable the thorough validation of the plant’s ethnopharmacological applications and validate its potential as a valuable resource for disease treatment and management in the future. Furthermore, standardized products or pure medicinal compounds can be easily evaluated and validated for safety.

## 5. Conclusions

*Parkia biglobosa* is an immensely beneficial plant from both a nutritional and traditional medicinal standpoint. The plant possesses antidiabetic, antihypertensive, hypolipidemic, neuroprotective, antimicrobial, and analgesic properties. However, the biochemical mechanisms behind these properties remain largely unexplored. The plant also offers a vast library of potential bioactive compounds with promising pharmacological activities. To advance our understanding, it is essential to focus research efforts on unraveling the structural and functional aspects of these phytochemicals that will establish the biochemical pathways responsible for the observed biological effects. Once identified, known constituents can be standardized and proper safety assessments performed to facilitate the advancement of these findings from the bench to the bedside.

## Figures and Tables

**Figure 1 ijerph-21-00394-f001:**
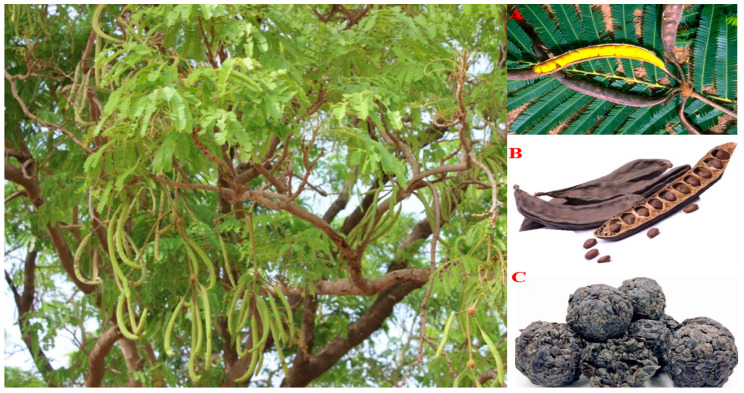
African locust bean tree (*Parkia biglobosa*): (**A**). African locust bean leaf with mature pod showing yellow pulp. (**B**). Dried pods and seeds. (**C**). Fermented, processed seeds (*Dawadawa*, *iru*, *soumbala*). Adapted from Pasiecznik [13].

## Data Availability

Not applicable.

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
