# Peer review of "Contemporary Insights into the Biological Mechanisms of Parkia biglobosa"

_ijerph, 2024, doi:10.3390/ijerph21040394_

Round 1
Reviewer 1 Report
Comments and Suggestions for Authors
The manuscript provides a comprehensive overview of the mechanisms of action of bioactive compounds of Parkia biglobosa supported by scientific research. However, there are some aspects that could be improved.
- Abstract
The overall abstract is concise and informative, but there is a need for revisions as follows:
The authors should add a brief of selection of the research works and database used in this review.
- Methodology
This manuscript lacks the methodology. So, the authors should outline the scope, database, inclusion and exclusion criteria, selection process, and data extraction.
- Ethnomedicinal Applications and Phytochemistry
It is recommended that the authors have to mention phytochemical ingredients in detail regarding extraction methods, isolation and analytical techniques.
Table 1 – it is informative, but it should include all extraction, isolation and analytical techniques.
- Pharmacological Activities and Biochemical Signaling
In general, there is a lack of specific details in mentioning the findings of previous studies such as the dose of the plant extract used in animal testing or the minimum inhibitory concentration (MIC) values in antimicrobial assays, which are essential to provide a comprehensive understanding of the methodologies employed and to facilitate critical evaluation, replication, and comparison of the results.
Sub-section - 3.3.1.1. Cytotoxic Activities of Essential Oils and Polyphenols
The authors are suggested to improve the explanation of a mechanism of action of plant bioactive compounds on the anti-microbial activity through previous studies related to bioactive compounds of the plant.
Here is the example of the published papers to support the explanation.
Natural phenolic compounds: Antimicrobial properties, antimicrobial mechanisms, and potential utilization in the preservation of aquatic products (https://doi.org/10.1016/j.foodchem.2023.138198).
Table 2 – it is informative, but it is suggested adding specific details of major findings. For example, the dose of the extracts used for animal testing, MIC values for antibacterial assays.
- Overall, there are some issues including missing italic format for scientific names, incorrect names of analytical techniques, and font size inconsistency.
Comments on the Quality of English LanguageSome typing and English grammatical errors are found in the manuscript. The authors should carefully check any errors.
Reviewer 2 Report
Comments and Suggestions for Authors
Overall quality and relevance of the manuscript
The manuscript provides a timely and relevant review on the current understanding of the biological activities and mechanisms of Parkia biglobosa, an important medicinal plant in Africa. It comprehensively covers the plant's ethnomedicinal uses, identified phytochemicals, pharmacological effects observed in preclinical studies, and the putative signaling pathways and mechanisms underlying these therapeutic actions. The review fills an important gap in collating the scattered data on this multipurpose plant and lays the groundwork to advance research on the characterization of its bioactive components and their mechanisms of action. The manuscript is well-structured and clearly written. The topic is appropriate for the scope of the International Journal of Environmental Research and Public Health.
Critical analysis
The literature review seems quite exhaustive, drawing from several preclinical studies on various crude extracts and isolates from different parts of Parkia biglobosa. The authors make a cogent case for the need to identify, standardize, and characterize the specific bioactive agents responsible for the observed medicinal properties as a prerequisite for clinical evaluations and pharmacological applications. The description of the mechanisms and pathways involved, though limited by the paucity of studies, provides useful insights and context. Areas that could benefit from expansion or further elaboration include:
- More details on the specific polyphenols, terpenoids, or other agents suspected to be key active components behind certain effects.
- Pharmacokinetic data if available, noting the bioavailability or metabolic fate of identified chemicals.
- Structure-activity relationship data for purified chemicals where research allows.
- Further exploration into the antimicrobial mechanisms which focused narrowly on essential oils and polyphenols.
Additional minor edits needed:
- Standardize font style/size, spacing, paragraph breaks
- Improve clarity on sub-headings and transitions between topics
- Check for redundancy between sections
- Correct minor typos, grammatical errors
In summary, this is a well-written review analyzing the current state of knowledge on an important medicinal plant. It could be improved by addressing the suggestions above and tightened further with careful editing. Overall, the manuscript merits indexation pending revisions.
Comments on the Quality of English LanguageNil
